# Illness perspectives in patients with primary aldosteronism

Oskar Ragnarsson [1,2] *, Andreas Muth [3,4], Gudmundur Johannsson [1,2], Eleftheria Gkaniatsa [1,2], Eva Jakobsson Ung [5,6], Sofie Jakobsson [5,6]

1 Department of Endocrinology, Sahlgrenska University Hospital, Gothenburg, Sweden, 2 Department of Internal Medicine & Clinical Nutrition, Institute of Medicine, Sahlgrenska Academy, University of Gothenburg, Gothenburg, Sweden, 3 Department of Surgery, Sahlgrenska University Hospital, Gothenburg, Sweden, 4 Department of Surgery, Institute of Clinical Sciences, Sahlgrenska Academy, University of Gothenburg, Gothenburg, Sweden, 5 Institute of Health and Care Sciences, Sahlgrenska Academy at the University of Gothenburg, Gothenburg, Sweden, 6 Centre for Person-Centred Care (GPCC), University of Gothenburg, Gothenburg, Sweden

* oskar.ragnarsson@medic.gu.se

**Data Availability Statement:** The raw data are recordings from interviews that cannot be made publicly available due to ethical restrictions, i.e. data contain potentially identifying or sensitive patient information. However, upon a reasonable

## Abstract

### Objective

The burden of symptoms and treatment in patients with primary aldosteronism (PA), as well as the patients' experience of the health care is sparsely studied. The objectives of this study were to describe symptoms considered to be the most troublesome by patients with PA, and to explore health related worries and expectations following treatment.

### Methods

This was an explorative qualitative study where 25 patients with PA, diagnosed between 2017 and 2019, were included; 13 patients who had undergone adrenalectomy and 12 who were receiving medical treatment. Data was collected during six group interviews and analyzed using a thematic approach.

### Results

Three main themes were identified: 1) *Distress of the past*, where the most important issues were struggle to receive a correct diagnosis, impaired well-being and the consumption of a large number of tablets, 2) *Satisfaction after receiving a correct diagnosis*, both in patients with unilateral and bilateral disease, but also dissatisfaction with lack of information about the disease, and 3) *Future concerns*, where worries about the long-term effects of PA on health in general dominated.

### Conclusions

Our findings illustrate several important issues related to PA where improvements in patient care are needed, including actions aiming at shortening the long diagnostic delay, a thorough information to the patients about the disease is of great importance, and that all patients with PA, regardless of treatment, would benefit from a structured long-term follow-up.

request, the raw data can be obtained by contacting the corresponding author (oskar. ragnarsson@medic.gu.se), or the head of the department of endocrinology, Lena Bokemark (lena.bokemark@vgregion.se).

**Funding:** The authors received no specific funding for this work.

**Competing interests:** The authors have declared that no competing interests exist.

## Introduction

Primary aldosteronism (PA) is caused by excess aldosterone production from one or both adrenal glands. PA is a common cause of secondary hypertension with an estimated prevalence of 5–13% among patients with high blood pressure [1]. Without disease-specific treatment, patients with PA have an increased risk of several vascular comorbidities, including atrial fibrillation, stroke, myocardial infarction and impaired renal function [2].

Approximately half of all patients with PA have a dominant aldosterone production from one of the adrenal glands, while the other half have a bilateral disease. Unilateral adrenalectomy is standard of care for patients with unilateral disease while medical treatment with mineralocorticoid receptor antagonist is the treatment of choice for patients with bilateral disease. Both surgical and medical treatments result in good blood pressure control, and the risk of future cardiovascular complications is decreased [3]. Whether medical treatment is as good for long-term outcome as surgical treatment for patients with PA is still unclear [4].

The knowledge of the impact of PA on quality of life, the burden of symptoms and treatment, and patient experience of the health care is limited. Some studies have shown that patients with PA have an impaired quality of life that improves by treatment [5, 6]. There are some indications that the improvement in quality of life is better in patients receiving surgical rather than medical treatment [6]. The reason for this may have several explanations, such as a better treatment effect with surgery as well as the feeling of being cured by surgery, as compared with life-long medical treatment. The mineralocorticoid receptor is abundantly expressed in the amygdala suggesting an important functional role in cognition [7] and animal studies suggest that activation of the renin-aldosterone-angiotensin system (RAAS) impairs cognitive function [8]. Furthermore, in patients with essential hypertension higher plasma aldosterone concentration has been associated with reduced cognitive function [9]. In patients with PA, however, this association has not been confirmed [10]. Thus, the impaired quality of life in patients with PA is, therefore, probably related to other factors such as the hypertension per se, and the large number of medications often needed for adequate control of the hypertension with high risk of troublesome side-effects.

The main purpose of this qualitative study was to explore which symptoms related to PA are considered to be the most bothersome by the patients. A further aim was to explore health related worries and expectations following treatment. The ultimate goal with the study was to identify factors of importance for patients with PA, and to be able to use these as outcome variables in future studies where different treatment alternatives for PA are evaluated.

## Methods

### Design

This was an explorative qualitative study where data collection and data analysis followed a thematic analysis design. The qualitative methodology was chosen since we considered it the best approach to describe the patient's experiences of being diagnosed and treated for PA.

### Data collection

Totally 63 patients diagnosed with PA who underwent adrenal vein catheterization at our institution between 2017 and 2019 were invited to participate in the study. Of these, 31 had undergone adrenalectomy due to unilateral PA and 32 patients with bilateral PA had received medical treatment.

The inclusion criteria were 1) age >18 years, 2) treatment with adrenalectomy, or initiation of treatment with mineralocorticoid receptor antagonist, at least one year before inclusion,

and 3) proficiency in the Swedish language. Surgically treated patients were identified through the Scandinavian Quality Register for Thyroid, Parathyroid and Adrenal Surgery, and medically treated patients by a search for the diagnostic code for PA (ICD-10 code E26.0) in the diagnosis-related group (DRG) registry at our hospital. Of 63 patients, 28 agreed to participate in the study, and 25 patients attended.

Between April and June 2021 six group interviews with 25 patients, 10 women and 15 men, were performed online via a videoconferencing platform. Median age of the patients was 56 years (range 37–69 years; Table 1). All 25 patients had initially been screened by measuring aldosterone and renin in plasma. Subsequently, PA was confirmed by using intravenous 4-hr saline infusion test in supine position where p-aldosterone >138 pmol/L was considered to be compatible with PA. Thereafter, all patients underwent continuous cosyntropin-stimulated sequential adrenal venous sampling (AVS). Adequate cannulation was checked by calculating selectivity index and lateralization by calculating lateralization index (LI) as previously described [11]. LI >4.0 was considered to be compatible with unilateral PA and LI <3.0 with bilateral disease. LI between 3 and 4 was considered to indicate unilateral disease if the contralateral index (aldosterone/cortisol nondominant adrenal vein)/(aldosterone/cortisol peripheral vein) was below 1.0 [11]. Based on the AVS results, 13 of 25 patients had unilateral PA and had undergone adrenalectomy and 12 were considered to have bilateral PA and were receiving medical treatment. All 13 patients who underwent adrenalectomy had complete biochemical success according to the PASO criteria [12], 6 had complete clinical success and 7 had partial clinical success. All 12 patients with bilateral disease were receiving treatment with mineralocorticoid receptor antagonist, 10 with eplerenone and two with spironolactone.

Three to five patients joined each interview, which were held by two researchers (EJU and SJ). One researcher moderated the interviews, and the other observed, took notes and controlled technical issues such as the audio digital recording. The aim with the interviews was to give each patient an opportunity to freely talk about his or her experiences of being diagnosed and treated for PA. The conversations focused on experienced symptoms, expectations before and after treatment, worries, and other important matters, as judged by the patients. Each participant had the opportunity to tell their story in relation to each subject and the moderator

**Table 1. Baseline characteristics of the study cohort.**

|  | All (n = 25) | Unilateral PA (n = 13) | Bilateral PA (n = 12) |
|---|---|---|---|
| **Sex** |  |  |  |
| Women | 10 | 5 | 5 |
| Men | 15 | 8 | 7 |
| **Age (years)** | 53 ± 10 | 53 ± 11 | 52 ± 9 |
| **Duration of hypertension (yrs)** | 6 (2–15) | 6 (3–15) | 5 (2–17) |
| **Antihypertensive drugs, no** | 2 (2–3) | 2 (2–3) | 2 (1–3) |
| **Antihypertensive drugs, DDD** | 3 (2–6) | 4 (2–6) | 2.5 (1–6) |
| **Potassium supplementation, n (%)** | 14 (56) | 10 (77) | 4 (33) |
| **Potassium supplementation (g/day)** | 1.9 (0–7.1) | 4.5 (0.4–12.5) | 0 (0–3) |
| **P-aldosterone (pmol/L)** | 461 (268–652) | 641 (451–931) | 289 (185–388) |
| **P-aldosterone post SIT (pmol/L)** | 384 (255–719) | 672 (417–802) | 270 (200–320) |
| **Duration of follow-up after treatment (months)** | 6 (4–33) | 4 (3.5–5.5) | 24 (12–38) |
| **Time from treatment to study inclusion (months)** | 30 ± 8 | 29 ± 7 | 33 ± 8 |

Data are presented as mean ± standard deviation or median (interquartile range). Abbreviations: Blood pressure, BP; Daily defined dose, DDD; primary aldosteronism, PA; saline infusion test, SIT

confirmed and asked for clarification when needed. The patients were encouraged to give examples and to probe their experiences. All six interviews lasted approximately 60 minutes.

### Data analysis

In line with the description of thematic analysis of Braun and Clarke [13], a first step in the analysis included reading all the interviews to grasp an understanding of the data. Analysis then proceeded with sorting and coding of data based on text segments. The understanding of the data was discussed within the research group and a search for patterns and potential themes was performed. The data analysis included a back-and-forth process between data, memos from the interviews and conceptualizing themes. Main themes and sub themes were gradually described and defined in more detail. Finally, three main themes and seven sub themes were identified.

### Ethical considerations

The study was conducted according to the declaration of Helsinki and was approved by the Regional Ethics Committee of West Sweden, Gothenburg, Sweden on February 17, 2021 (DNR 2021–00195). All patients received written and verbal information about the study and gave written consent to participate.

## Results

When the patients talked about their experiences of PA, the narratives moved between the past, the present and the future. Descriptions from living with PA moved back and forth between the illness trajectory in the past, as they remember it, the present as they experience it, and the future as they imagine it. The analysis revealed three main themes: *Distress of the past* which included symptoms and signs and the consequences of PA for health, the large number of medications, and the struggle to receive adequate care; *Satisfaction after receiving a correct diagnosis* that comprised both reconciliation with the illness despite being bilateral, and the joy of being diagnosed with unilateral disease and cured following surgical treatment, as well as ideas about the cause of the illness, and health-promotive activities; *Future Concerns* that included potential harmful effects of the disease on the body, and concerns whether their children may have inherited the disease (Fig 1).

### Distress of the past–Symptoms and signs

All patients had a history of high blood pressure for several years, ranging from 1–25 years. Many times, it was difficult for the patients to remember when the high blood pressure was first discovered. One patient expressed: *I have had high blood pressure for as long as I can remember*. The narratives contained various descriptions of symptoms and signs that had often been present for years. The patients related the symptoms and signs to different causes; their high blood pressure, low potassium concentration and/or side effects from the medications. Several patients referred to a general and unspecific experience of "not feeling well". This was expressed by one patient as: *It felt like I was not happy, and I also had a general strange feeling in the body*. Another patient described it as: *It felt like the whole body was out of order*. The most common symptom was headache. The headache was described as intense and pulsating, got worse after physical activity, and was not relieved with painkillers. Arthralgias, generalized bodily pain, muscle cramps, fatigue and decreased exercise capacity were also common symptoms. Other less frequently mentioned symptoms were urinary frequency,

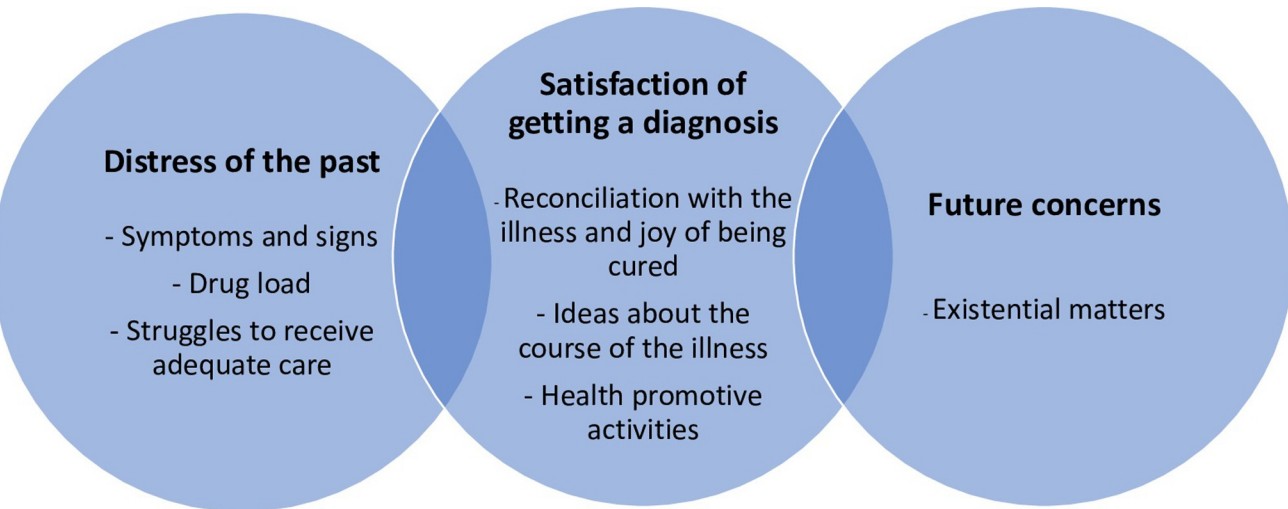

**Fig 1. The main themes of illness perspectives in patients with primary aldosteronism.**

swollen legs, dizziness, weight changes (both weight loss and weight gain), impatience, irritability, feeling worried and being sad.

## Distress of the past–Drug load

The time before PA was diagnosed was characterized by many patients as a time of frequent, and often unsuccessful, attempts to lower the blood pressure by using many different antihypertensives, and a large number of tablets, including potassium supplementation. Patients gave several examples of how the type and dose of medications were changed back and forth, both due to inadequate effect on the blood pressure as well as intolerance. Taking many tablets was perceived as a sign that something was seriously wrong in the body, and was associated with an increased risk for developing even more serious diseases in the future. Taking a large number of tablets was generally considered to be unhealthy, time consuming and troublesome when being away from home. The large number of tablets were described by some as "*a whole meal*".

## Distress of the past–Struggles to receive adequate care

The majority of the patients described a lack of continuity with a frequent change of physicians. Receiving adequate care was experienced as a struggle, characterized by lack of ambitions by the health care providers to identify the cause for the high blood pressure, difficulties in getting contact with them, as well as inadequate follow-up after investigations and/or initiation of treatment. The patients strived for investigations that would identify the cause of their high blood pressure, instead of only treating the hypertension. The narratives described a feeling of not being taken seriously, not being listened to, or abandoned: *It feels like that you are not been taken seriously. I had a physician who told me that I was making up my symptoms.* The patients had received several different explanations for their symptoms, including that the symptoms were caused by diabetes, menopause, migraine, overweight, and not exercising enough. The high blood pressure was sometimes considered to be related to temporary stress when visiting a physician, or related to a stressful life in general. Only a few patients had a positive experience of continuity in health care where PA was suspected instantly when their high blood pressure was noted. One of these patients described her good relationship with her

physician: *I have been followed up ever since [I was diagnosed with PA] so I think we are on the right track, no surgery though. Still following up on me and hoping the blood pressure goes down. So that´s where we are.*

The patients perceived that the physicians in specialized care units and physicians in primary care did not agree on which investigations were necessary, the same routine blood samplings were frequently repeated instead of doing more specific investigations to diagnose the cause of the symptoms. It was also a common experience that physicians from different health care units did not talk to each other, and were reluctant to change each other's prescriptions of medication. Instead of changing medications, new medications were added. The lack of continuity was often considered to have caused a long diagnostic delay. Nevertheless, PA was most often first suspected by the patients' ordinary general practitioners, although sometimes by a temporary substitute: *Yes, it simply feels like a roulette, it's all about seeing the right physician who really does something.* In few cases the specific test used to diagnose PA was initiated by the patients themselves or because a relative demanded it. In a small number of patients, a more serious events such as a heart attack, led to the diagnosis of PA.

## Satisfaction of getting a diagnosis–Reconciliation with the illness and joy of being cured

Relief and gratitude for having received a correct diagnosis was strongly expressed by almost all of the patients. Patients with bilateral disease had to reconcile with the fact that the disease is incurable and some described a disappointment of having a chronic illness. Nevertheless, for patients with bilateral disease it felt good to take a disease-specific medical treatment instead of unspecific treatment only based on a symptom. One man said: *That there was no surgery is understandable since both adrenal glands were affected. And now the values are good. I'm still taking blood pressure medication. Both my parents took blood pressure medication, and they were active as long as they lived, so I can live with that. Right now, I feel satisfied.* Receiving a correct diagnosis and adequate treatment also provided an explanation for the often "difficult-to-understand symptoms" in the past. However, some patients expressed that it was difficult to find an adequate dose and some experienced side effects that affected quality of life and/or exercise capacity. One man said: *The body does not tolerate physical exercise in the same way as before I became ill. It pushes me down and prevents me from maximizing myself.* Another side effect of the medications was leg cramps.

Patients with unilateral disease who have undergone adrenalectomy were very happy to have been cured. It was expressed as a great advantage to have escaped all the medications, medications that the patients expressed great antipathy to. One patient expressed his joy by saying: *I think it is fantastic that one day you eat nine tablets and the next [following surgery] you do not need a single tablet. So, it was such a striking difference. Just after the adrenal gland was removed, the problem did not exist. I did not think that the body could ever really react so quickly to such a thing [the surgery].* Several of these patients expressed that they no longer thought about their illness or that they were ill. However, the diagnosis was shocking for some: *For me it was quite shocking, so it is still a chronic thing, it is a tumor. For the patient, it [receiving the diagnosis] is tough and not just a relief.*

## Satisfaction of getting a diagnosis–Ideas about the cause of the illness

Many patients reflected on the cause of the disease. A common notion was that stress had triggered the disease or that one had acquired it in another way. One man says: *I've had quite a stressful time at work, I think it's a big culprit in the whole thing.* Another notion was that the disease might be hereditary. Several patients had parents with hypertension and were

considering whether it was PA that the parents actually had. One patient said: *Both my parents had high blood pressure periodically. Therefore, it is probably genetic*. A prevailing notion was that the disease was "an error in the system". A patient described: *You completely sabotage the hormonal system in the body, for a long time. It probably takes a very long time to recover, but the question is whether it will ever do so.*

## Satisfaction of getting a diagnosis–Health-promotive activities

The patients described that they practiced a number of different health-promotive activities such as taking their own blood pressure, exercising various forms of physical activity, eating healthy food and above all reduced stress in everyday life. One patient illustrated this: *Right now, I live very healthy I think, I have lost 13 kg since last winter, am physically active with paddle and golf. So, I think I'm having a good life right now*. These health-promotive activities were seen as an investment in health. There was a great awareness of the importance of staying fit. For patients with bilateral disease, the motive for staying fit were to be in as little need as possible of medications. Several patients wanted to lose weight. Others controlled their diet for other reasons: *I am just trying to cut down on salt intake. I know that it causes the blood pressure to go up. I am having another problem with my work because we are having lunch at work, and most of the food they are serving is very salty*. Patients checked their own blood pressure at home and sometimes several times a day. A patient who regularly checked his blood pressure at home said: *I was told to check my blood pressure regularly and I cannot run to a health center every other day*. However, measuring own blood pressure could also have negative consequences as all the measurements could lead to exaggerated meticulousness and an urge to constantly check and have control over the blood pressure.

Despite all health-promotive activities, it was obvious that the patients lacked knowledge about their disease. It was not enough to only attend to health care appointments, they needed specific information about PA. A patient said: *You get a disease that you have never heard of. The first thing I thought was what an adrenal gland is*. Because the disease was unusual, you do not meet others who suffer from the same disease and the possibility of peer support was described as limited. Another patient with unilateral disease, and who underwent surgery, illustrated the problem with health care professionals: *There was quite a bit of information to take in both before and after the surgery; this is how it can go; this is how you can feel. The staff did not have an experience [of caring for patients with primary aldosteronism]. It's understandable, but I experienced it quite clearly*. A patient with bilateral disease and who was checked at an endocrine specialist clinic said: *I can say that when I was diagnosed 4 years ago, I did not know where it [the adrenal gland] was, and I have not received any further information. You read a lot online, but when you ask questions, there is almost no knowledge*. A common notion was that patients did not receive any answers or concrete advice from the health care professionals on how to live with the disease: *Questions such as how to manage your life, what can happen in the future and what to be observant of are often dismissed with: It's OK, you can live as usual.*

## Concerns of the future–Existential matters

When patients thought about the future there were some concerns. Several patients took up heredity and expressed concerns about that their children and grandchildren may carry the disease. A common concern was that the disease, the high blood pressure, may have caused irreversible damage to the body such as atherosclerosis and/or increased the risk for a heart attack. A patient with unilateral disease said: *On the one hand, I can worry that it has worn on the body [the high blood pressure] and on the other hand I can worry that something happens to*

*the adrenal gland that I have left, it feels like you are more vulnerable, so I can worry about that.* Some patients had deep existential thoughts about longevity. One woman said: *It is not relevant for me to be 90 years old.* Another woman said: *I think I will die prematurely.* A younger woman worried about her children: *It is a lot that bothers me about this disease because I feel I have to be there for the kids because they need me.* Some patients with bilateral disease were concerned about the side effects of the medications, while patients with unilateral PA were worried about development of another or recurrent disease in the remaining adrenal gland in the future.

## Discussion

To our knowledge, this is the first study that has used qualitative methodology to investigate illness perspectives in patients with PA. By interviewing patients that had either been operated or where on targeted medical treatment for PA, we identified three main themes; distress before the disease was diagnosed, satisfaction after receiving the correct diagnosis and future concerns. Our findings are novel and brings new disease related issues into the light, issues that are apparently of great importance for patients with PA and that can be used to improve the care of patients with PA.

The symptoms described in retrospect from the patients before the diagnosis, as well as the experience of receiving the correct treatment, show that the burden of the disease was considerable before the diagnosis. The feeling of being sick was to some degree related to the large number of tablets needed to treat the hypertension, and in many cases to correct the hypokalemia. However, despite all these medications the situation was not controlled. The feeling of relief was considerable when the correct diagnosis was made and a specific treatment could be given.

Another observation of interest was how many patients had a general feeling of being unwell, including the presence of bothersome symptoms such as headache, arthralgias, generalized pain, muscle cramps, fatigue, irritability, worries and sadness. This agrees with studies showing that patients with PA have worse quality-of-life compared to the general population [6, 14, 15], and that anxiety, irritable mood, high stress level, psychological distress, depression and nervousness are more common in patients with PA than healthy controls [15, 16]. It can be speculated that some of these symptoms may be related to the effects of a long-standing overproduction of aldosterone on the brain [8], while others are more likely to be related to chronic hypokalemia and adverse effects from the blood pressure lowering treatment [17]. Symptoms such as headache, dizziness, restlessness and fatigue are also common in patients with essential hypertension [18]. In fact, patients with essential hypertension have comparable impairments in quality of life as patients with PA who have not yet received disease-specific treatment, supporting that uncontrolled hypertension and the burdens of intensive antihypertensive treatment is of major importance for general well-being [19].

The time before PA was diagnosed was difficult for many patients. In many cases the patients experienced that they had to fight with the health care system to get the tests that are necessary to diagnose the disease, and that it took long time to get the adequate testing. This is in line with our recent finding, showing that even though the detection rate of PA among patients with hypertension is increasing, it is still greatly underdiagnosed [20]. Similar results have been found in studies from Germany, Italy and USA, showing that patients with hypertension who have a relatively high risk of having PA are infrequently tested [21–23]. The reason for this is probably multifactorial. However, the cumbersome diagnostic process, including biochemical screening, then confirmatory testing, and finally, adrenal vein sampling, is undoubtedly one of the main reasons. Another reason is probably the lack of continuity that

many of the patients in the current study described, i.e. Swedish patients often meet several different physicians when they visit the primary health care instead of meeting the same doctor regularly. In fact, continuity of care is associated with better health-related outcomes, greater patient satisfaction and is more cost-effective [24].

The patients in this study have all been screened for PA, undergone confirmatory testing, subtype evaluation and have received disease-specific treatment with numerous contacts with general and specialized care. Still, the information provided about PA was considered unsatisfactory by many patients. Important questions for the patients such as if the disease was hereditary, which lifestyle changes should be adopted, or potential future consequences of the disease and treatment remained. Some advice given, *i.e.* lack of restrictions after minimally invasive surgery—from the surgeon´s point of view an attractive quality–was perceived as dismissive or ignorant. Differing perceptions of given information and emotional support between patients and health care professionals has been demonstrated earlier [25, 26]. Targeted interventions to increase patient's knowledge of their disease and treatment in chronic illness have been shown to improve patient´s health literacy and adherence to treatment [27], hopefully also increasing patient satisfaction. The current study suggests that a better understanding between patients and caregivers is needed, and that improved patient health literacy and satisfaction can be achieved by highlighting pertinent themes and issues to be addressed in the care of patients with PA.

The study design enabled an exploration of PA-related lived experiences from the patients' point of view which would have been difficult to investigate with quantitative methodology. The heterogenous group of patients in terms of laterality (bi- and unilateral PA), gender and age distribution and varying duration with undiagnosed PA provided a broad range of narratives for the analysis. We choose to include patients with both bi- and unilateral disease in the group interviews, even if this could potentially inhibit the narratives from one of the treatment groups. However, the group discussions were dialectic, both within and across the diagnosis, and patients agreed and/or disagreed with each other, and gave various examples of experience that were reflected with the discussion. Several patients had similar experiences despite having a bi- or unilateral PA, especially experiences regarding aspects described in the themes "distress of the past" and "future concerns". At the same time, the heterogenous group can be considered a limitation, and subgroup analyses comparing e.g. surgically treated patients with complete, partial or absent success, and medically treated patients receiving spironolactone versus eplerenone, cannot be performed due to the relatively small number of patients included in the study.

The videoconferencing platform was selected as patients were spread over a large geographical area and to secure a safe data collection during the Covid-19 pandemic. One patient, however, who was interested in the study could not participate since he lacked access to necessary technical equipment. Both the interviewers and the remaining participants were familiar with the equipment required for participating in the videoconferencing platform. The varying backgrounds and preunderstanding of the researchers permitted a wide interpretation of data that enhances the process of analysis and per se the trustworthiness of the result.

## Conclusions

With this study we provide new information on illness perspectives in patients with PA. By using a qualitative study design, three main themes were identified: 1) *Distress of the past*, where struggle to receive a correct diagnosis, impaired well-being and the consumption of a large number of tablets, were the most important issues 2) *Satisfaction after receiving a correct diagnosis*, both in patients with unilateral and bilateral disease, but also dissatisfaction with

lack of information about the disease, and 3) *Future concerns*, where worries about the long-term effects of PA on health in general, as well as whether relatives may inherit the disease, were common. These results illustrate several important aspects of PA, pointing to that actions are needed aiming at shortening the long diagnostic delay of the disease, a thorough information to the patients about the disease is fundamental for acceptance and coping, and a structured long-term follow-up is necessary for all patients that have received treatment for PA.

## Acknowledgments

We thank Annika Reibring and Anna Cederberg-Olsson for their excellent work in transcibing the interviews, and Maria Nilsson, registered nurse, for help with administrative issues.

## Author Contributions

**Conceptualization:** Oskar Ragnarsson, Andreas Muth, Gudmundur Johannsson, Eleftheria Gkaniatsa, Eva Jakobsson Ung, Sofie Jakobsson.

**Data curation:** Eva Jakobsson Ung, Sofie Jakobsson.

**Formal analysis:** Eva Jakobsson Ung, Sofie Jakobsson.

**Investigation:** Eleftheria Gkaniatsa, Eva Jakobsson Ung, Sofie Jakobsson.

**Methodology:** Eva Jakobsson Ung, Sofie Jakobsson.

**Supervision:** Oskar Ragnarsson, Sofie Jakobsson.

**Validation:** Eleftheria Gkaniatsa.

**Writing – original draft:** Oskar Ragnarsson, Andreas Muth, Gudmundur Johannsson, Eva Jakobsson Ung, Sofie Jakobsson.

**Writing – review & editing:** Eleftheria Gkaniatsa.

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
