## [Decision Letter · Decision Letter 0]

15 Aug 2022

PONE-D-22-08400

Illness perspectives in patients with primary aldosteronism

PLOS ONE

Dear Dr. Ragnarsson,

Thank you for submitting your manuscript to PLOS ONE. After careful consideration, we feel that it has merit but does not fully meet PLOS ONE’s publication criteria as it currently stands. Therefore, we invite you to submit a revised version of the manuscript that addresses the points raised during the review process.

Please note that we have only been able to secure a single reviewer to assess your manuscript. We are issuing a decision on your manuscript at this point to prevent further delays in the evaluation of your manuscript. Please be aware that the editor who handles your revised manuscript might find it necessary to invite additional reviewers to assess this work once the revised manuscript is submitted. However, we will aim to proceed on the basis of this single review if possible. 

The reviewer was overall positive about your manuscript, but had feedback about reporting in the Methods, Results, and Discussion sections. Among their concerns, the reviewer expressed that the article should discuss how results for different groups of participants compared with one another, i.e. differences between treatment groups, and differences between participants who showed complete, partial, or no clinical success following adrenalectomy. Please address all of the reviewer's comments in manuscript revisions and your Response to Reviewer letter.

We look forward to receiving your revised manuscript.

Kind regards,

Renee Hoch, Ph.D.

Managing Editor, PLOS Publication Ethics

PLOS ONE

https://journals.plos.org/plosone/s/file?id=ba62/PLOSOne_formatting_sample_title_authors_affiliations.pdf".

2. Please note that in order to use the direct billing option the corresponding author must be affiliated with the chosen institute. Please either amend your manuscript to change the affiliation or corresponding author, or email us at plosone@plos.org with a request to remove this option.

Reviewers' comments:

Reviewer's Responses to Questions

**Comments to the Author**

1. Is the manuscript technically sound, and do the data support the conclusions?

Reviewer #1: Yes

2. Has the statistical analysis been performed appropriately and rigorously? 

Reviewer #1: N/A

3. Have the authors made all data underlying the findings in their manuscript fully available?

Reviewer #1: No

4. Is the manuscript presented in an intelligible fashion and written in standard English?

Reviewer #1: Yes

5. Review Comments to the Author

Reviewer #1: The study of Ragnarsson O. and colleagues qualitatively describes the quality of life of patients with primary aldosteronism before, during and after treatment in both, unilateral and bilateral disease. This is the first study to adopt a qualitative approach to this topic and therefore is of particular interest.

The study is well designed and described; however, some issues are presents and are listed below:

Methods:

• I suggest to provide more information for screening test, confirmatory test and respective cut-offs for PA diagnosis.

• More details on AVS methods (stimulated or unstimulated? simultaneous or sequential?) criteria should be provided

Results:

• How many medically treated patients were treated with spironolactone? How many with eplerenone? Did the authors notice any difference between the two groups?

• There was any difference between patients that showed complete clinical success after adrenalectomy (defined according to PASO criteria, Williams TA, Lancet DE 2017 doi: 10.1016/

S2213-8587(17)30135-3) and those that showed partial or absent clinical success? Did biochemical success influence the satisfaction after adrenalectomy?

Discussion:

• Many studies reported that QOL is reduced in patients with PA, compared to the general population. However, patients with PA show similar QOL compared to patients with EH matched for hypertension severity (Buffolo F, EJCI 2021, DOI: 10.1111/eci.13419). This finding may suggest that hypertension and antihypertensive drug treatment may be the most important factors affecting QOL in PA patients, more than the effect of aldosterone per se. I suggest to briefly discuss this aspect in Discussion.

• Some authors reported that patients with PA showed reduced well-being and higher levels of stress and psychological distress (Sonino N, JCEM 2011; doi: 10.1210/jc.2010-2723). Moerover, depressive symptoms improve at FU after adrenalectomy (Citton, BMC surgery 2019, doi:10.1186/s12893-018-0432-1). I would discuss this themes in Discussion describing whether some of this aspects emerged from the interviews of this study

6. PLOS authors have the option to publish the peer review history of their article (what does this mean?). If published, this will include your full peer review and any attached files.

Reviewer #1: No

---

## [Author Response · Author response to Decision Letter 0]

28 Sep 2022

Comments to the Author

Reviewer #1: The study of Ragnarsson O. and colleagues qualitatively describes the quality of life of patients with primary aldosteronism before, during and after treatment in both, unilateral and bilateral disease. This is the first study to adopt a qualitative approach to this topic and therefore is of particular interest. 

The study is well designed and described; however, some issues are presents and are listed below:

Methods:

• I suggest to provide more information for screening test, confirmatory test and respective cut-offs for PA diagnosis.

We have added this information to the revised version as suggested by the reviewer.

• More details on AVS methods (stimulated or unstimulated? simultaneous or sequential?) criteria should be provided

We have added this information to the revised version as suggested by the reviewer.

Results:

• How many medically treated patients were treated with spironolactone? How many with eplerenone? Did the authors notice any difference between the two groups?

Of 12 patients with bilateral disease, 10 were receiving eplerenone and two spironolactone. This information has been added to the method-section. Unfortunately, the low number of patients on spironolactone makes it impossible to compare the two groups.

• There was any difference between patients that showed complete clinical success after adrenalectomy (defined according to PASO criteria, Williams TA, Lancet DE 2017 doi: 10.1016/S2213-8587(17)30135-3) and those that showed partial or absent clinical success? Did biochemical success influence the satisfaction after adrenalectomy?

Information on clinical and biochemical success in surgically treated patients, according to PASO, has been added to the method-section. The question whether there is a difference between patients with complete versus partial/absent clinical success is interesting. However, due to the relatively low number of patients (7 versus 6), this type of analysis cannot be performed in this study. We have, however, added a brief text to the discussion on this limitation.

Discussion:

• Many studies reported that QOL is reduced in patients with PA, compared to the general population. However, patients with PA show similar QOL compared to patients with EH matched for hypertension severity (Buffolo F, EJCI 2021, DOI: 10.1111/eci.13419). This finding may suggest that hypertension and antihypertensive drug treatment may be the most important factors affecting QOL in PA patients, more than the effect of aldosterone per se. I suggest to briefly discuss this aspect in Discussion.

We thank the reviewer for this comment. We have added a brief discussion on this important topic to the discussion.

• Some authors reported that patients with PA showed reduced well-being and higher levels of stress and psychological distress (Sonino N, JCEM 2011; doi: 10.1210/jc.2010-2723). Moerover, depressive symptoms improve at FU after adrenalectomy (Citton, BMC surgery 2019, doi:10.1186/s12893-018-0432-1). I would discuss this themes in Discussion describing whether some of this aspects emerged from the interviews of this study.

We appreciate this comment and have added discussion on this topics to the revised version of the manuscript.

---

## [Decision Letter · Decision Letter 1]

6 Nov 2022

Illness perspectives in patients with primary aldosteronism

PONE-D-22-08400R1

Dear Dr. Ragnarsson,

We’re pleased to inform you that your manuscript has been judged scientifically suitable for publication and will be formally accepted for publication once it meets all outstanding technical requirements.

Kind regards,

Esmat Mehrabi

Academic Editor

PLOS ONE

---

## [Editor Report · Acceptance letter]

15 Nov 2022

PONE-D-22-08400R1 

Illness perspectives in patients with primary aldosteronism 

Dear Dr. Ragnarsson:

I'm pleased to inform you that your manuscript has been deemed suitable for publication in PLOS ONE. Congratulations! Your manuscript is now with our production department. 

Kind regards, 

on behalf of

Dr. Esmat Mehrabi 

Academic Editor

PLOS ONE